# Colorectal Cancer in the Young: Research in Early Age Colorectal Cancer Trends (REACCT) Collaborative

**DOI:** 10.3390/cancers15112979

**Published:** 2023-05-30

**Authors:** Alexandra M. Zaborowski

**Affiliations:** Centre for Colorectal Disease, St. Vincent’s University Hospital, Elm Park, D04 T6F4 Dublin, Ireland; zaborowa@tcd.ie; Tel.: +353-877783672

**Keywords:** early-age onset, colorectal cancer

## Abstract

**Simple Summary:**

Colorectal cancer is becoming more common in adults under 50. This trend has been observed worldwide, the reasons for which are unclear. Research has suggested that young patients present with more advanced diseases compared to their older counterparts yet have similar survival. How young patients respond to current treatments is unknown as they represent a small proportion of studies to date. The aim of this study is to evaluate disease-specific features and survival of patients diagnosed with colorectal cancer under 50 years of age. Increased understanding of the characteristics of early-age onset colorectal cancer will help to design age-specific preventative, diagnostic, and treatment strategies for these patients and to inform public health policies and initiatives.

**Abstract:**

**Background:** The incidence of colorectal cancer (CRC) is increasing in the young (under 50). Defining the clinicopathological features and cancer-specific outcomes of patients with early-onset CRC is important to optimize screening and treatment strategies. This study evaluated disease-specific features and oncological outcomes of patients with early-onset CRC. **Methods:** Anonymized data from an international collaboration were analyzed. The inclusion criteria for this study were patients aged <50 years with stage I-III disease surgically resected with curative intent. Overall and disease-free survival were calculated using the Kaplan–Meier method. **Results:** A total of 3378 patients were included, with a median age of 43 (18–49) and a slight male preponderance (54.3%). One-third had a family history of colorectal cancer. Almost all (>95%) of patients were symptomatic at diagnosis. The majority (70.1%) of tumors were distal to the descending colon. Approximately 40% were node positive. Microsatellite instability was demonstrated in one in five patients, representing 10% of rectal and 27% of colon cancers. A defined inherited syndrome was diagnosed in one-third of those with microsatellite instability. Rectal cancer displayed a worse prognosis stage for stage. Five-year disease-free survival for stage I, II, and III colon cancer was 96%, 91%, and 68%, respectively. The equivalent rates for rectal cancer were 91%, 81%, and 62%. **Conclusions and relevance:** The majority of EOCRC would be captured with flexible sigmoidoscopy. Extending screening to young adults and public health education initiatives are potential interventions to improve survivorship.

## 1. Introduction

Although the overall incidence and mortality of colorectal cancer (CRC) have decreased globally since the introduction of population-based screening, there has been a significant rise in incidence among adults aged less than 50 years. Early age onset CRC (EOCRC) now represents the second most common cancer and the third leading cause of cancer-related death in this age group [1]. Similar trends have been observed in Europe and North America, with the greatest increase in distal colon and rectal cancers [2,3,4,5]. Analysis of 143 million people from 20 European countries showed CRC incidence has increased by 7.9% per year among subjects aged 20–29 years, by 4.9% among those aged 30–39 years, and by 1.6% among those aged 40–49 years from 2004 to 2016 [2]. Based on current data, it is estimated that within the next decade, 1 in 4 rectal cancers and 1 in 10 colon cancers will be diagnosed in individuals under 50 [4].

The factors driving these escalating trends are unclear. Although more likely to occur in the context of a hereditary cancer syndrome, the majority of EOCRCs are sporadic [6,7,8]. Environmental and lifestyle factors such as diet, cigarette smoking, and alcohol consumption in isolation do not explain the observed trends. Importantly, EOCRC represents a global phenomenon and affects both genders. Potential risk factors hypothesized include obesity, infections, antibiotics, and alterations to the gut microbiome [9].

The clinicopathological characteristics of EOCRC differ from those of late-onset disease. EOCRC typically involves the distal colon or rectum, presents at an advanced disease stage, and displays unfavorable histological features such as poor differentiation, mucin, and signet-ring morphology [10,11,12]. These differences may be associated with different survival. Survival data for young patients are lacking and conflicting, and despite increased use of neoadjuvant/adjuvant treatment among patients with EOCRC sensitivity to conventional chemo(radio)therapy is unknown [12,13,14]. Unique tumor biology may influence response to treatment, and age at diagnosis is not considered in modern therapeutic strategies. The long-term effects of chemoradiotherapy on quality of life and fertility are of particular importance in this patient group. The potential overtreatment of patients with low-risk diseases must be questioned in the absence of a definitive oncological benefit.

Although increasing in incidence, EOCRC accounts for a very small proportion of CRCs overall. Individual institutional data alone are of limited value in addressing the big questions that surround this disease. Whilst a huge body of data on older patients with CRC exists, data relating to young patients is relatively lacking. Thus, the Research in Early Age Colorectal Cancer Trends (REACCT) Collaborative was established to combine data from institutions worldwide. This study aimed to evaluate the disease-specific features as well as oncological outcomes of patients diagnosed with CRC aged less than 50 years.

## 2. Methods

### 2.1. Study Participants

A retrospective international multicentre observational study to assess the clinicopathological features and oncological outcomes of patients diagnosed with EOCRC was performed. Patients were identified from the REACCT Collaborative database. Inclusion criteria were adults aged between 18 and 49 years with a histologically confirmed diagnosis of adenocarcinoma of the colon or rectum who underwent surgery with curative intent. Rarer carcinomas and tumors of the appendix or anus were excluded. Patients with distant metastases at diagnosis were excluded.

### 2.2. Data Collection

Each participating institution is a tertiary referral center with expertise in the management of CRC. Data entry, storage, and high-end data protection were via the REDCap system. The data retrieved were fully anonymized. Ethical approval was sought at an individual institutional level. Collected data included patient demographics, neoadjuvant therapy, surgical intervention, histopathological features, surgical outcomes, adjuvant therapy, and cancer-specific as well as overall survival information. Rectal cancer and tumors of the colon were evaluated separately. Rectal cancer was defined as tumors within 15 cm from the anal verge on colonoscopy. Clinical staging was according to the 8th edition of the American Joint Committee on Cancer (AJCC) tumor node metastasis (TNM) staging system. Chemotherapy was administered according to institutional protocol and the multidisciplinary team. Details pertaining to the decision-making or whether patients were enrolled in a clinical trial were not recorded. Microsatellite instability was identified by PCR or by loss of mismatch repair proteins, MLH1, PMS2, MSH2, and MSH6 on immunohistochemistry. The term MSI is used throughout for consistency. The definition of hereditary cancer syndrome was a diagnosis of a constitutive pathogenic variant on germline testing.

### 2.3. Study Endpoints

The primary endpoints were overall and disease-free survival. Secondary endpoints for rectal cancer were pathological response rates and the impact of neoadjuvant and adjuvant therapy on survival. For colon cancer, the secondary endpoint was the impact of adjuvant therapy on survival.

### 2.4. Statistical Analysis

Continuous variables were described as mean (±standard deviation) or median values (range) and, depending on their distribution, were compared by the Student *t*-test or Mann–Whitney U test. Categorical variables were reported as percentages. The χ^2^ test or Fisher exact test were used to assess the association of categorical variables. The Kaplan–Meier method was used to calculate survival statistics. Independent variables were entered into a multivariable binary logistic regression model. Variables that were found to be significant at univariable analysis (*p*-value < 0.05) were entered into the multivariable model. A significance level of 0.05 was used for all analyses; reported *p*-values are 2-tailed. Data were analyzed using SPSS^®^ software version 24.0 (IBM, Armonk, NY, USA).

## 3. Results

### 3.1. Baseline Demographics

A total of 3,378 patients diagnosed with non-metastatic colorectal cancer under the age of 50 were included in the study. The median (range) age was 43 (18–49) years, and 1835 (54.3%) were male. Median (range) BMI was 24 (12–59). The majority of patients were white Caucasian (45.2%) or Asian (29.6%). Never smokers accounted for 67%, and 15% had a history of excess alcohol consumption defined as >14 standard units/week. Most patients were well with an ASA grade of 1 or 2 (92.4%) and an ECOG performance status of 0 or 1 (98.1%). Only 3.2% (*n* = 109) had a diagnosis of inflammatory bowel disease, of which ulcerative colitis was the most common. Almost one-third (31.8%) had a first-degree relative with CRC. No patients had a previous diagnosis of CRC. The majority of patients were European or North American. Baseline demographics are summarised in Table 1.

### 3.2. Clinical Characteristics

Rectal tumors accounted for 44.2% (*n* = 1494), while 1884 patients (55.8%) had tumors located in the colon. The majority of colonic tumors involved the sigmoid colon (*n* = 674, 35.8%). Synchronous tumors were uncommon (0.8%). Most patients were symptomatic at diagnosis. The diagnosis was an incidental finding in only 184 patients (5.4%). Clinical characteristics are summarised in Table 1.

### 3.3. Neoadjuvant and Adjuvant Therapy

Neoadjuvant (chemo)radiotherapy was administered to 527 (35.3%) patients with rectal cancer. The median radiotherapy dose was 50Gy, and the most common chemotherapy agent administered was capecitabine. Neoadjuvant chemotherapy was administered to 58 (3.0%) patients with colon cancer. Adjuvant chemotherapy was given to 821 (43.6%) and 615 (41.2%) patients with colon and rectal cancer, respectively. FOLFOX and CAPOX were the most commonly administered regimes, summarised in Table 2. The criteria for administration of adjuvant chemotherapy are unknown. It is presumed that the presence of high-risk pathological features would have guided decision-making.

### 3.4. Pathology

Overall, an R0 resection was achieved in 3126 (92.5%) patients: 1728 colonic and 1398 rectal. Lymphovascular invasion, extramural invasion, and perineural invasion were present in 34.6%, 29.9%, and 19.4% of colon cancers and 33.0%, 22.6%, and 21.1% of rectal cancers, respectively. A pCR was identified in 171 (26.8%) patients with rectal cancer who received neoadjuvant therapy.

### 3.5. Molecular Features

MSI status was known in 1033 patients (611 colonic and 422 rectal). MSI was identified in 165 (27.0%) patients with colon cancer and 46 (10.9%) with rectal cancer. A definable hereditary cancer syndrome was diagnosed in 60 (36.4%) and 11 (24.0%) patients with MSI colon and rectal cancer, respectively. Notably, 58 (35.2%) of those with MSI colon cancer and 14 (30.4%) with MSI rectal cancer had not had genetic testing at the time of data collection.

### 3.6. Oncological Outcomes

#### 3.6.1. Colon Cancer

The 5-year OS rates were 99%, 94%, and 82% for stages I, II, and III, respectively. The 5-year DFS rates were 96%, 91%, and 68% for stages I, II, and III. Disease recurrence occurred in 247 patients (17.9%); 56 (4.1%) and 191 (13.8%) developed locoregional and systemic recurrence.

#### 3.6.2. Rectal Cancer

The 5-year OS rates were 98%, 90%, and 76% for stages I, II, and III, respectively. The 5-year DFS rates were 91%, 81%, and 62% for stages I, II, and III. Disease recurrence occurred in 265 patients (21.2%); 83 (6.6%) developed locoregional recurrence, and 212 (17.0%) developed systemic recurrence.

### 3.7. Factors Influencing Survival

R0 resection and pN0 status were significantly associated with better DFS survival in both colon and rectal cancer. Univariable and multivariable analyses are summarised in Table 3.

### 3.8. Impact of Neoadjuvant and Adjuvant Therapy on Survival

Neoadjuvant therapy did not improve DFS in patients with pathological (yP) node-negative (log-rank 0.92, *p* = 0.338) or positive rectal cancer (indeed, it was associated with worse disease-free survival in the latter; log-rank 11.95, *p* = 0.001). This was largely due to persistently positive disease despite receiving up-front therapy (i.e., poor tumor regression). DFS was modestly better in those patients who received adjuvant chemotherapy in stage III colon cancer (log-rank 2.84, *p* = 0.092); a larger sample size would be needed to achieve statistical significance. Meanwhile, the need for adjuvant chemotherapy in stage II colon cancer was associated with worse DFS (log-rank 6.02, *p* = 0.014). In rectal cancer, patients who received adjuvant chemotherapy, similar DFS was observed for stage II (log-rank 0.56, *p* = 0.760) and III (log-rank 1.08, *p* = 0.299) disease.

## 4. Discussion

The epidemiology of CRC has changed significantly over the past three decades. Many questions exist surrounding the aetiology, molecular profile, and genetic component of the disease, and a large volume of data specific to this patient group are lacking. This study presents the clinicopathological features and oncological outcomes of 3378 patients diagnosed with non-metastatic CRC aged less than 50 from all over the world. The majority of patients were European or North American. Japan, Australia, and New Zealand were also well represented.

In the present series, there was a modest male preponderance, most patients had a normal BMI, and one-third had a family history of CRC. In keeping with previously published data, the majority of tumors (70%) were located in the sigmoid colon and rectum [4,15,16,17]. Why young adults are disproportionately affected by a distal colon and rectal cancer is unclear, and deciphering the aetiology of rectal versus colon cancer will be key to developing preventative strategies [18,19]. Importantly, tumor location guides optimum screening methods. Flexible sigmoidoscopy should capture the majority of EOCRCs, representing a potential screening tool with family history-based risk assessment as an adjunct.

EOCRC is more frequently associated with adverse histopathological features and more advanced disease at diagnosis relative to late-onset disease [10,12,20]. In this study, the proportions of patients with lymphovascular invasion, extramural venous invasion, perineural invasion, and node-positive disease were higher compared to those reported in older patients [21,22,23,24]. Advanced disease stage may reflect aggressive tumor biology and/or be due to a delay from symptom onset to diagnosis, either patient or physician related [25,26]. The vast majority of patients were symptomatic, highlighting the importance of educational initiatives to raise awareness among young adults, primary care physicians, and clinicians and ensure timely diagnosis and intervention.

Survival data for patients with EOCRC are conflicting. Despite advanced disease stage and unfavorable histopathological features, some studies report equivalent oncological outcomes, whilst others report better survival among young patients [22,27,28,29,30,31]. In the present study, rectal cancer displayed a worse prognosis with 5-year DFS rates of 91%, 81%, and 62% for stages I, II, and III, respectively, compared to 96%, 91%, and 68% for those with colon cancer. These survival data are comparable to those of stage-matched older patients. Considerable heterogeneity in survival has been observed in those under 50, supporting further age-based subgrouping in future studies [32].

Young patients are more likely to receive neoadjuvant and adjuvant therapy and to receive multi-agent regimes at all stages compared to their older counterparts, despite no significant gain in adjusted survival [12,14,33,34]. They are also more likely to receive systemic therapy outside National Comprehensive Cancer Network treatment guidelines [12,13,28,29]. In the present series, just over one in three patients with stage I/II rectal cancer received neoadjuvant therapy, whilst one-third of patients with stage II colon or rectal cancer received adjuvant chemotherapy.

The sensitivity of EOCRC to conventional chemotherapy is unclear, as young patients account for a small proportion of clinical trials. Adjuvant chemotherapy was associated with a modest improvement in DFS among patients with stage III colon cancer in the present series. A larger sample size would be needed to demonstrate statistical significance. Interestingly, worse DFS was observed in patients with stage II disease who received adjuvant chemotherapy, suggesting the presence of unfavorable histopathological features. Similarly, patients with stage III rectal cancer who received neoadjuvant therapy had worse DFS than those who did not. Alternative or adjunctive oncotherapeutic strategies such as total neoadjuvant therapy or immunotherapy may optimize survivorship [35,36].

The estimated prevalence of hereditary cancer syndromes in EOCRC ranges between 5 and 35%, compared to 2–5% of CRCs overall [6,7,8]. In the present study, MSI was identified as 1 in 5 patients (27% of colon cancers and 10% of rectal cancers). A hereditary cancer syndrome was diagnosed in just over 1 in 3 patients with colon cancer and 1 in 4 with rectal cancer, highlighting the importance of reflex genetic testing in patients with MSI EOCRC. The therapeutic implication of MSI is the potential for immunotherapy with checkpoint blockade, with remarkable success observed in the neoadjuvant setting [36,37,38].

Although genetic predisposition plays an important role in EOCRC, it must be acknowledged that the majority of patients have sporadic disease [39,40]. Thus, screening patients with a family history or known genetic predisposition will only capture a small proportion of patients with EOCRC. Furthermore, the majority of patients are symptomatic at diagnosis. As symptoms such as abdominal pain, rectal bleeding, etc., frequently overlap with benign conditions, public health initiatives are needed to raise awareness and educate both physicians and patients about the risk of CRC in young adults. Screening with flexible sigmoidoscopy would identify the majority of patients who develop EOCRC.

Limitations of this study include the retrospective nature, data heterogeneity, and variable treatment algorithms across the collaborative group. However, this study presents real-world data relating to a disease that has been, until recently, relatively uncommon in the young age group. The patients in the present study were not directly compared to older patients, as a large volume of evidence for older patients already exists. Future studies should focus on deciphering the optimum oncotherapeutic management of patients with EOCRC.

## 5. Conclusions

In this multicenter study of 3378 patients with EOCRC, the majority had tumors involving the sigmoid colon or rectum. Patients with rectal cancer displayed worse five-year disease-free survival stage for stage. MSI was present in 20.0% (1 in 4 colon cancers and 1 in 10 rectal cancers). Approximately one-third of patients with MSI tumors (colonic and rectal) were diagnosed with a definable hereditary cancer syndrome. Public health initiatives are important to ensure early diagnosis and optimize survivorship.

## Figures and Tables

**Table 1 cancers-15-02979-t001:** Demographics and clinicopathological data.

	Colon N = 1884	Rectal N = 1494
Median age (range)	43 (18–49)	43 (18–49)
Male	997 (52.9%)	838 (56.1%)
Median BMI (range)	24 (13–59)	23 (12–50)
Inflammatory Bowel Disease	67 (3.6%)	42 (2.8%)
First Degree Relative with CRC	591 (31.4%)	482 (32.3%)
Known hereditary cancer syndrome	0 (0%)	0 (0%)
Symptoms		
Abdominal pain	580 (30.8%)	174 (11.6%)
Change in bowel habit	393 (20.9%)	380 (25.4%)
Rectal bleeding	535 (28.4%)	873 (58.4%)
Anaemia	235 (12.5%)	51 (3.4%)
Incidental finding	132 (7.0%)	52 (3.5%)
Other	170 (9.0%)	88 (5.9%)
Tumor site		
Rectosigmoid junction	202 (10.7%)	-
Sigmoid colon	674 (35.8%)	-
Descending colon	160 (8.5%)	-
Splenic flexure	121 (6.4%)	-
Transverse	191 (10.1%)	-
Hepatic flexure	58 (3.1%)	-
Ascending colon	252 (13.4%)	-
Caecum	239 (12.7%)	-
Synchronous tumor	13 (0.6%)	14 (0.9%)
Neoadjuvant therapy	58 (3.0%)	527 (35.3%)
Type of neoadjuvant therapy		
Chemoradiotherapy	-	434 (29.0%)
Radiotherapy only	-	59 (4.0%)
Chemotherapy only	58 (3.0%)	34 (2.3%)
pTNM stage		
I	448 (23.8%)	460 (30.8%)
II	626 (32.2%)	411 (27.5%)
III	810 (43.0%)	623 (41.7%)

**Table 2 cancers-15-02979-t002:** Neoadjuvant and adjuvant therapy according to disease stage.

	Stage I	Stage II	Stage III
Rectal cancer			
Neoadjuvant (chemo)radiotherapy	160 (34.8%)	173 (42.1%)	194 (31.1%)
Adjuvant chemotherapy	70 (15.2%)	126 (30.7%)	419 (67.3%)
Colon cancer			
Adjuvant chemotherapy	9 (2.0%)	215 (34.3%)	597 (73.7%)

**Table 3 cancers-15-02979-t003:** (**a**) Univariable and multivariable logistic regression of factors predicting disease-free survival in patients with colon cancer I–III. (**b**) Univariable and multivariable logistic regression of factors predicting disease-free survival in patients with rectal cancer stage I–III.

(a)						
	Univariable			Multivariable		
Variable	HR	95% CI	*p*	HR	95% CI	*p*
Age (years)	1.007	0.986, 1.028	0.522	-	-	-
Male	0.750	0.567, 0.989	0.041	-	-	-
cTNM stage I–II	2.524	1.717, 3.710	<0.0001	-	-	-
cTNM stage III	0.396	0.270, 0.582	<0.0001	-	-	-
R0 resection	3.647	2.144, 6.203	<0.0001	3.681	1.640, 8.263	0.002
pN0 status	5.084	3.723, 6.942	<0.0001	3.209	1.835, 5.611	<0.0001
MSI	0.610	0.350, 1.062	0.081	-	-	-
(**b**)						
	**Univariable**			**Multivariable**		
**Variable**	**HR**	**95% CI**	** *p* **	**HR**	**95% CI**	** *p* **
Age (years)	0.986	0.967, 1.006	0.173	-	-	-
Male	0.918	0.705, 1.195	0.525	-	-	-
cTNM stage I–II	2.263	1.263, 1.643	<0.0001	-	-	-
cTNM stage III	0.399	0.287, 0.555	<0.0001	-	-	-
R0 resection	1.001	1.007, 1.013	0.048	1.016	1.001, 1.030	0.036
pN0 status	2.749	2.092, 3.614	<0.0001	3.187	1.660–6.121	<0.0001
MSI	0.375	0.143, 0.984	0.046	-	-	-

## Data Availability

The data that support the findings of this study are available from the corresponding author upon reasonable request.

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
