# Peer review of "Colorectal Cancer in the Young: Research in Early Age Colorectal Cancer Trends (REACCT) Collaborative"

_cancers, 2023, doi:10.3390/cancers15112979_

Round 1
Reviewer 1 Report
The authors hope to define characteristics of colon adenocarcinoma in patients under 50 that are different from patients older than 50, but fail to query their database for the over 50 cohort. In other words, they could have compared their own patients over 50 to patients under 50, and not used historical controls. Had they done so, they could answer the potential critique that the differences they found were not due to age, but were due to their unique patient population.
The REACCT Collaboration should be described in greater detail. It is unclear what the geographic distribution of the centers is, whether all patients are included or just a sampling, how the disease free survival data is collected, and exactly what is meant by CRC, since colon cancer is a broader term than adenocarcinoma. Also, the methods mention institutional databases and independent review, making it unclear if this is a centralized database in which all contributors use the same definitions and format.
Table 2 adds little information to the text and can be eliminated.
Is the limited number of patients who had MSI testing due to the study being based on patients enrolled prior to widespread screening or is it due to only a fraction of patients being chosen for screening basd on clinical criteria. The manuscript refers to MSI testing, which is not commonly done. Is that what the authors mean, or are the referring to MMR testing.
The disease free survival would be better demonstrated with a Kaplan- Meier curve, with the number of patients at risk displayed on the x-axis at yearly intervals. This would give readers a better sense for the length of follow-up and is a more intuitive way of presenting the data.
Author Response
The authors hope to define characteristics of colon adenocarcinoma in patients under 50 that are different from patients older than 50, but fail to query their database for the over 50 cohort. In other words, they could have compared their own patients over 50 to patients under 50, and not used historical controls. Had they done so, they could answer the potential critique that the differences they found were not due to age, but were due to their unique patient population.
There is a huge body of data for older patients with CRC, the focus of this study was to aggregate data on young patients which is relatively lacking.
The REACCT Collaboration should be described in greater detail. It is unclear what the geographic distribution of the centers is, whether all patients are included or just a sampling, how the disease free survival data is collected, and exactly what is meant by CRC, since colon cancer is a broader term than adenocarcinoma. Also, the methods mention institutional databases and independent review, making it unclear if this is a centralized database in which all contributors use the same definitions and format.
Centres from all over the world were included. The majority of patients were European and North American. Parts of Asia (e.g. Japan), Australia/New Zealand and South America were represented. Africa, India and China were underrepresented. All patients with non-metastatic disease entered into the database were included. DFS was calculated from date of surgery to date of recurrence or date of last known follow up. Inclusion criteria included adenocarcinoma of the colon and rectum. Rarer carcinomas were excluded. Tumours of the appendix or anus were excluded.
Table 2 adds little information to the text and can be eliminated.
The authors felt this table presented the table in a manner that was easy for the reader to digest. It can be removed at the editor’s discretion.
Is the limited number of patients who had MSI testing due to the study being based on patients enrolled prior to widespread screening or is it due to only a fraction of patients being chosen for screening basd on clinical criteria. The manuscript refers to MSI testing, which is not commonly done. Is that what the authors mean, or are the referring to MMR testing.
Microsatellite instability was identified by PCR or by loss of mismatch repair proteins, MLH1, PMS2, MSH2, and MSH6 on immunohistochemistry. The term MSI was used in the text for consistency. The definition of hereditary cancer syndrome was diagnosis of a constitutive pathogenic variant on germline testing. The study interval is long (20 years). Not all institutions were reflexly testing MSI status.
The disease free survival would be better demonstrated with a Kaplan- Meier curve, with the number of patients at risk displayed on the x-axis at yearly intervals. This would give readers a better sense for the length of follow-up and is a more intuitive way of presenting the data.
A KM curve was felt to be unnecessary as the survival of two or more groups of patients are not being compared. The absolute values (1, 3 and 5 year DFS) are of more interest to readers. However, graphs can be provided if the editor wishes.
Reviewer 2 Report
This study included totally 3,378 patients underwent colectomy with a median age of 43. They found the majority (70.1%) tumors were distal to the descending colon. However, these findings, similar to previous reports, highlighted Left-Sided dominance of early-onset colorectal cancers, for examples, Diseases of the Colon & Rectum 62(8):p 920-924, August 2019; Dis Colon Rectum. 2018;61:897–902 ; and Diseases of the Colon & Rectum 46(7):p 904-910, July 2003. Segev L,et al therefore suggested that a rationale for screening flexible sigmoidoscopy in the young. Dis Colon Rectum. 2018;61:897–902.
A big confusing exists in the text between abstract, line 19…. A defined inherited syndrome was diagnosed in one third of those…..and Table 1 : the item “ Known hereditary cancer syndrome” is 0(0%). It needs to be clarified.
Author Response
This study included totally 3,378 patients underwent colectomy with a median age of 43. They found the majority (70.1%) tumors were distal to the descending colon. However, these findings, similar to previous reports, highlighted Left-Sided dominance of early-onset colorectal cancers, for examples, Diseases of the Colon & Rectum 62(8):p 920-924, August 2019; Dis Colon Rectum. 2018;61:897–902 ; and Diseases of the Colon & Rectum 46(7):p 904-910, July 2003. Segev L,et al therefore suggested that a rationale for screening flexible sigmoidoscopy in the young. Dis Colon Rectum. 2018;61:897–902.
A big confusing exists in the text between abstract, line 19…. A defined inherited syndrome was diagnosed in one third of those…..and Table 1 : the item “ Known hereditary cancer syndrome” is 0(0%). It needs to be clarified.
Known hereditary syndrome refers to a hereditary cancer syndrome already diagnosed prior to diagnosis of CRC. No patients had a known hereditary syndrome prior to diagnosis.
Reviewer 3 Report
The manuscript describing the characteristics, aetiological factors, and oncological outcome of colorectal cancers in individuals under 50 years of age as a result of the REACCT international collaboration, is a very valuable study as it includes a large number of patients.
However, I have some comments on the paper:
1. The communication is very short and cannot be published as an 'article'. It should either be changed to communication' or expanded to a word count of 4000 words.
2. If such a large number of patients' data has been collected, why is there no mention of the types of screening tests used and their specificity and sensitivity in relation to the final diagnosis, in addition to the patients' symptoms and complaints?
3. The histological diagnosis of tumors should also be reported, not only the molecular pathological results. Were there rarer types of carcinoma in the study population, or were there tumors with complex tissue structures?
Otherwise, the results reported speak for themselves, and I found nothing to complain about.
Major revision is recommended.
Author Response
The manuscript describing the characteristics, aetiological factors, and oncological outcome of colorectal cancers in individuals under 50 years of age as a result of the REACCT international collaboration, is a very valuable study as it includes a large number of patients.
However, I have some comments on the paper:
1. The communication is very short and cannot be published as an 'article'. It should either be changed to communication' or expanded to a word count of 4000 words.
The authors feel the purpose is to present the data and discuss rather than to reach a specific word count or be repetitious. Publication as a short report or communication would also be appropriate if the editors wish.
2. If such a large number of patients' data has been collected, why is there no mention of the types of screening tests used and their specificity and sensitivity in relation to the final diagnosis, in addition to the patients' symptoms and complaints?
Data pertaining to screening tests are not available unfortunately. Patients' symptoms are presented in Table 1.
3. The histological diagnosis of tumors should also be reported, not only the molecular pathological results. Were there rarer types of carcinoma in the study population, or were there tumors with complex tissue structures?
Otherwise, the results reported speak for themselves, and I found nothing to complain about.
One of the inclusion criteria was adenocarcinoma of the colon or rectum. Rarer carcinoma types were excluded as were tumours of the appendix and anus.
Reviewer 4 Report
Early Onset CRC is an extremely relevant subject and the authors describe a large multicentric series, adding important data to the literature. However, some points need clarification:
- What was the geographic distribution of patients/centers? There are geographic diferences in EOCRC's incidence and some areas may have been over or under represented in the sample.
- How many of the patients had previous cancers and how many had received prior chemo and/or radiotherapy? These are also known risk factors for EOCRC.
- Why did some patients with colon cancer receive neoadjuvant chemotherapy? If they were all stage I-III, this is not standard therapy. Were they in a clinical trial? And why did some stage I patients with colon cancer receive adjuvant chemotherapy?
- Which criteria were used to select stage II colon cancer patients for adjuvant chemotherapy?
- How was MSI status assessed? Was MMR protein expression available? Were MSI-H patients tested for BRAF mutation? In patients with MLH!/PMS2 loss of expression and BRAF mutation, genetic testing may not be indicated.
- In the «Discussion»: the sentence «Alternative or adjunctive oncotherapeutic strategies such as TNT or immunotherapy may optimize survivorship.» is speculative and unsupported by these study's results.
- The EOCRC patients' data was compared to historical data and not to the older population followed in the same centers in the same period, therefore we cannot be certain that the described characteristics are age specific. - this should be discussed.
Minor comments:
- Which AJCC edition was used in staging?
- Conflicts of interests should be stated for all authors, not just the first.
Author Response
Q1. What was the geographic distribution of patients/centers? There are geographic diferences in EOCRC's incidence and some areas may have been over or under represented in the sample.
The study involved centres from all over the world. The majority of patients were European or North American. Parts of Asia (Japan) and Australia/New Zealand were well represented. Africa, India and China were underrepresented.
Q2. How many of the patients had previous cancers and how many had received prior chemo and/or radiotherapy? These are also known risk factors for EOCRC.
These data are unknown.
Q3. Why did some patients with colon cancer receive neoadjuvant chemotherapy? If they were all stage I-III, this is not standard therapy. Were they in a clinical trial? And why did some stage I patients with colon cancer receive adjuvant chemotherapy?
Chemotherapy was given according to local institutional protocol. Reasons for administration were not accessible. It is unknown if these patients were part of a clinical trial. Published data suggest patients with EOCRC are treated outside of current NCCN guidelines more frequently than their older counterparts. This likely reflects the idea EOCRC is an aggressive disease that should be treated aggressively.
Q4. Which criteria were used to select stage II colon cancer patients for adjuvant chemotherapy?
Administration of chemotherapy was according to the MDT and institutional protocol. Reasons for administration were not accessible. It is presumed high risk pathological features would have been part of the criteria.
Q5. How was MSI status assessed? Was MMR protein expression available? Were MSI-H patients tested for BRAF mutation? In patients with MLH!/PMS2 loss of expression and BRAF mutation, genetic testing may not be indicated.
Microsatellite instability was identified by PCR or by loss of mismatch repair proteins, MLH1, PMS2, MSH2, and MSH6 on immunohistochemistry. For the purpose of clarity the term MSI was used describe both in the results section. BRAF status was not available for the majority of patients unfortunately, one of the limitations of retrospective analyses.
Q6. In the «Discussion»: the sentence «Alternative or adjunctive oncotherapeutic strategies such as TNT or immunotherapy may optimize survivorship.» is speculative and unsupported by these study's results.
The authors feel that emphasis on oncotherapeutic strategies is important in future research in EOCRC. Immunotherapy with checkpoint blockade has been shown to be successful in the neoadjuvant setting in MSI colon cancer.
Q7. The EOCRC patients' data was compared to historical data and not to the older population followed in the same centers in the same period, therefore we cannot be certain that the described characteristics are age specific. - this should be discussed.
A huge body of data already exists for older patients with CRC. The focus of this study was to aggregate data for young patients which is relatively lacking.
Minor comments:
Which AJCC edition was used in staging?
The 8th Edition
Conflicts of interests should be stated for all authors, not just the first.
This statement refers to all authors.
Reviewer 5 Report
thank you for giving me the opportunity to review this large series of young patients with colorectal cancer. in order to homogenize the series, i will remove the patients with ulcerative colitis because the natural history of colorectal cancer is not the same.
how can it be explained that less than one third of patients with stage III rectal cancer had only neoadjuvant treatment?
Why does neoadjuvant treatment in rectal cancer have a negative impact on recurrence-free survival?
Author Response
Q.1 How can it be explained that less than one third of patients with stage III rectal cancer had only neoadjuvant treatment?
The authors hypothesise that significant variations in MDT decision making have historically existed in the management of patients with early age onset colorectal cancer. Some patients have been overtreated whilst others may have been undertreated. It is possible physicians (and/or the patients themselves) wished to avoid the consequences of chemoradiotherapy (e.g. infertility, sexual/urinary dysfunction).
Q2. Why does neoadjuvant treatment in rectal cancer have a negative impact on recurrence-free survival?
It is possible that worse recurrence-free survival observed in patients who received neoadjuvant therapy is reflective of unfavorable histopathological features and persistant disease (poor tumour regression).
Round 2
Reviewer 1 Report
My major criticism was the failure to compare young patients in the database to older patients in the database, instead of comparing young patients to historical controls. However, if the database only includes young people, as appears to be the case, then that criticism cannot realistically be addressed. My other criticisms have been addressed.
Reviewer 3 Report
The authors revised their work according to the suggestions of all Reviewers.
The revised version can be accepted for publication.
Reviewer 4 Report
Most of the major issues remained unanswered and weren't even discussed as limitations in the reviewed manuscript.
